# HyP-NeRF: Learning Improved NeRF Priors using a HyperNetwork

**Bipasha Sen***
MIT CSAIL
bise@mit.edu

**Gaurav Singh***
IIIT, Hyderabad
gaurav.si[†]

**Aditya Agarwal***
MIT CSAIL
adityaag@mit.edu

**Rohith Agaram**
IIIT, Hyderabad
rohith.agaram[†]

**K Madhava Krishna**
IIIT, Hyderabad
mkrishna@iiit.ac.in

**Srinath Sridhar**
Brown University
srinath@brown.edu

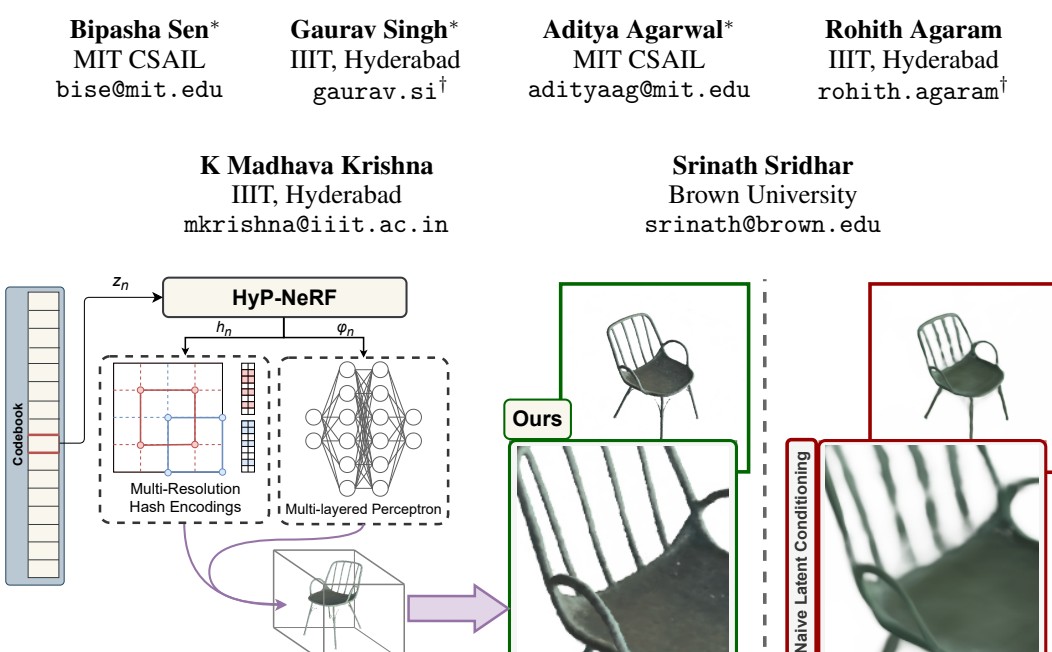

Figure 1: We propose HyP-NeRF, a latent conditioning method that learns improved quality NeRF priors using a hypernetwork to generate instance-specific multi-resolution hash encodings along with neural network weights. The figure showcases the fine details preserved in the NeRF generated by HyP-NeRF (green box) as opposed to the NeRF generated by naive conditioning (red box) in which, a hypernetwork predicts only the neural weights while relying on the standard positional encodings.

## Abstract

Neural Radiance Fields (NeRF) have become an increasingly popular representation to capture high-quality appearance and shape of scenes and objects. However, learning generalizable NeRF priors over categories of scenes or objects has been challenging due to the high dimensionality of network weight space. To address the limitations of existing work on generalization, multi-view consistency and to improve quality, we propose HyP-NeRF, a latent conditioning method for learning generalizable category-level NeRF priors using hypernetworks. Rather than using hypernetworks to estimate only the weights of a NeRF, we estimate both the weights and the multi-resolution hash encodings [35] resulting in significant quality gains. To improve quality even further, we incorporate a denoise and finetune strategy that denoises images rendered from NeRFs estimated by the hypernetwork and finetunes it while retaining multiview consistency. These improvements enable us to use HyP-NeRF as a generalizable prior for multiple downstream tasks including NeRF reconstruction from single-view or cluttered scenes, and text-to-NeRF. We provide

*Equal authors (order decided by a coin flip)
† @research.iiit.ac.in

37th Conference on Neural Information Processing Systems (NeurIPS 2023).

qualitative comparisons and evaluate HyP-NeRF on three tasks: generalization, compression, and retrieval, demonstrating our state-of-the-art results. [3]

# 1   Introduction

Neural fields, also known as implicit neural representations (INRs), are neural networks that learn a continuous representation of physical quantities such as shape or radiance at any given space-time coordinate [70]. Recent developments in neural fields have enabled significant advances in applications such as 3D shape generation [77], novel view synthesis [32, 2], 3D reconstruction [72, 66, 38, 68], and robotics [52, 51]. In particular, we are interested in Neural Radiance Fields (NeRF) that learn the parameters of a neural network $f_\phi(\mathbf{x}, \theta) = \{\sigma, c\}$, where $\mathbf{x}$ and $\theta$ are the location and viewing direction of a 3D point, respectively, and $\sigma$ and $c$ denote the density and color estimated by $f_\phi$ at that point. Once fully trained, $f_\phi$ can be used to render novel views of the 3D scene.

Despite their ability to model high-quality appearance, NeRFs cannot easily generalize to scenes or objects not seen during training thus limiting their broader application. Typically, achieving generalization involves learning a prior over a data source such as image, video, or point cloud distributions [21, 20, 58, 76, 29, 49], possibly belonging to a category of objects [65, 46]. However, NeRFs are continuous volumetric functions parameterized by tens of millions of parameters making it challenging to learn generalizable priors. Previous works try to address this challenge by relying on 2D image-based priors, 3D priors in voxelized space, or by using latent conditioning.

Image-based priors re-use the information learned by 2D convolutional networks [75, 34] but may lack 3D knowledge resulting in representations that are not always multiview consistent. Methods that learn 3D priors in voxelized space [33] suffer from high compute costs and inherently lower quality due to voxelization limitations. Latent conditioning methods [19, 43] learn a joint network $f(\mathbf{x}, \theta, z)$ where $z$ is the conditioning vector for a given object instance. These methods retain the advantages of native NeRF representations such as instance-level 3D and multiview consistency, but have limited capacity to model a diverse set of objects at high visual and geometric quality. InstantNGP [35] provides a way to improve quality and speed using *instance-specific* multi-resolution hash encodings (MRHE), however, this is limited to single instances.

We propose HyP-NeRF, a latent conditioning method for learning improved quality generalizable **category-level NeRF priors** using hypernetworks [15] (see Figure 1). We take inspiration from methods that use meta-learning to learn generalizable representations [55, 48] while retaining the quality of instance-specific methods [35]. Our hypernetwork is trained to generate the parameters–both the multi-resolution **hash encodings (MRHE) and weights**–of a NeRF model of a given category conditioned on an instance code $z_n$. For each instance code $z_n$ in the learned codebook, HyP-NeRF estimates $h_n$ denoting the instance-specific MRHE along with $\phi_n$ indicating the weights of an MLP. Our key insight is that estimating both the MRHEs and the weights results in a significant improvement in quality. To improve the quality even further, we denoise rendered views [42] from the estimated NeRF model, and finetune the NeRF with the denoised images to enforce multiview consistency. As shown in Figure 2 and the experiments section, this denoising and finetuning step significantly improves quality and fine details while retaining the original shape and appearance properties.

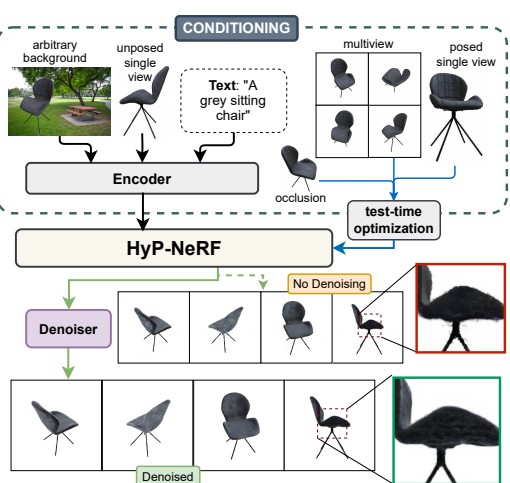

Figure 2: Once trained, HyP-NeRF acts as a prior to support multiple downstream applications, including NeRF reconstruction from single or multi-view images and clustered scene images, and text-to-NeRF. We further improve quality using our denoising network.

Once HyP-NeRF is trained, it can be used as a NeRF prior in a variety of different applications such as NeRF reconstruction from a single view posed or unposed images, single pass text-to-NeRF,

---

[3]Project page: hyp-nerf.github.io

or even the ability to reconstruct real-world objects in cluttered scene images (see Figure 2). We show qualitative results on applications and quantitatively evaluate HyP-NeRF's performance and suitability as a NeRF prior on the ABO dataset [11] across three tasks: generalization, compression, and retrieval. To sum up our contributions:

1. We introduce HyP-NeRF, a method for learning improved quality NeRF priors using a hypernetwork that estimates *instance-specific* hash encodings and MLP weights of a NeRF.
2. We propose a denoise and finetune strategy to further improve the quality while preserving the multiview consistency of the generated NeRF.
3. We demonstrate how our NeRF priors can be used in multiple downstream tasks including single-view NeRF reconstruction, text-to-NeRF, and reconstruction from cluttered scenes.

## 2   Related Work

**Neural Radiance Fields** [32] (NeRFs) are neural networks that capture a specific 3D scene or object given sufficient views from known poses. Numerous follow-up work (see [62, 70] for a more comprehensive review) has investigated improving quality and speed, relaxing assumptions, and building generalizable priors. Strategies for improving quality or speed include better sampling [2], supporting unbounded scenes [3], extensions to larger scenes [69, 60], using hybrid representations [35, 74], using learned initializations [61, 5, 45], or discarding neural networks completely [73, 59]. Other work relaxes assumption of known poses [67, 31, 27, 25, 9, 53], or reduce the number of views [75, 43, 4, 14, 34, 44, 71, 28, 37]. Specifically, PixelNeRF [75] uses convolution-based image features to learn priors enabling NeRF reconstruction from as few as a single image. VisionNeRF [28] extends PixelNeRF by augmenting the 2D priors with 3D representations learned using a transformer. Unlike these methods, we depend purely on priors learned by meta-learning, specifically by hypernetworks [15]. AutoRF [34] and LolNeRF [43] are related works that assume only a single view for each instance at the training time. FWD [5] optimizes NeRFs from sparse views in real-time and SRT [45] aims to generate NeRFs in a single forward pass. These methods produce NeRFs of lower quality and are not designed to be used as priors for various downstream tasks. In contrast, our focus is to generate high-quality multiview consistent NeRFs that capture fine shapes and textures details. HyP-NeRF can be used as a category-level prior for multiple downstream tasks including NeRF reconstruction from one or more posed or unposed images, text-to-NeRF (similar to [40, 18]), or reconstruction from cluttered scene images. Additionally, HyP-NeRF can estimate the NeRFs in a single forward pass with only a few iterations needed to improve the quality. Concurrent to our work, NerfDiff [13] and SSDNeRF [8] achieve high quality novel view synthesis by using diffusion models.

**Learning 3D Priors**. To learn category-level priors, methods like CodeNeRF [19] and LolNeRF [43] use a conditional NeRF on instance vectors $z$ given as $f(\mathrm{x}, \theta, z)$, where different $z$s result in different NeRFs. PixelNeRF [75] depends on 2D priors learned by 2D convolutional networks which could result in multi-view inconsistency. DiffRf [33] uses diffusion to learn a prior over voxelized radiance field. Like us, DiffRF can generate radiance fields from queries like text or images. However, it cannot be directly used for downstream tasks easily.

Our approach closely follows the line of work that aims to learn a prior over a 3D data distribution like signed distance fields [39], light field [55], and videos [48]. We use meta-learning, specifically hypernetworks [15], to learn a prior over the MRHEs and MLP weights of a fixed NeRF architecture. LearnedInit [61], also employs standard meta-learning algorithms for getting a good initialization of the NeRF parameters. However, unlike us, they do not use a hypernetwork, and use the meta-learning algorithms only for initializing a NeRF, which is further finetuned on the multiview images. Methods like GRAF [47], $\pi$-GAN [6], CIPS-3D [79], EG3D [7], and Pix2NeRF [4] use adversarial training setups with 2D discriminators resulting in 3D and multiview inconsistency. [40, 64, 16] tightly couple text and NeRF priors to generate and edit NeRFs based on text inputs. We, on the other hand, train a 3D prior on NeRFs and separately train a mapping network that maps text to HyP-NeRF's prior, decoupling the two.

## 3   HyP-NeRF: Learning Improved NeRF prior using a Hypernetwork

Our goal is to learn a generalizable NeRF prior for a category of objects while maintaining visual and geometric quality, and multiview consistency. We also want to demonstrate how this prior can be

used to enable downstream applications in single/few-image NeRF generation, text-to-NeRF, and reconstruction of real-world objects in cluttered scenes.

**Background**. We first provide a brief summary of hypernetworks and multi-resolution hash encodings that form the basis of HyP-NeRF. Hypernetworks are neural networks that were introduced as a meta-network to predict the weights for a second neural network. They have been widely used for diverse tasks, starting from representation learning for continuous signals [55, 54, 57, 48], compression [36, 12], few-shot learning [50, 23], continual learning [63]. Our key insight is to use hypernetworks to generate both the network weights and instance-specific MRHEs.

**Neural Radiance Fields** (NeRF) [32, 2] learn the parameters of a neural network $f_\phi(\mathbf{x}, \theta) = \{\sigma, c\}$, where $\mathbf{x}$ and $\theta$ are the location and viewing direction of a 3D point, respectively, and $\sigma$ and $c$ denote the density and color predicted by $f_\phi$ at that point. Once fully trained, $f_\phi$ can be used to render novel views of the 3D scene. NeRF introduced *positional encodings* of the input 3D coordinates, $\mathbf{x}$, to a higher dimensional space to capture high-frequency variations in color and geometry. InstantNGP [35] further extended this idea to *instance-specific* multi-resolution hash encodings (MRHE) to encode $\mathbf{x}$ dynamically based on scene properties. These MRHEs, $h$, are learned along with the MLP parameters, $\phi$ for a given NeRF function, $f$ and show improved quality and reduced training/inference time.

**Image Denoising** is the process of reducing the noise and improving the perceptual quality of images while preserving important structural details. Recent advancements in deep learning-based image restoration and denoising techniques [26, 24, 10] have demonstrated remarkable success in removing noise and enhancing the perceptual quality of noisy images that may have suffered degradation. Such networks are trained on large datasets of paired noisy and clean images to learn a mapping between the degraded input and the corresponding high-quality output by minimizing the difference between the restored and the ground truth clean image. In our case, we use denoising to improve the quality of our NeRF renderings by reducing artifacts and improving the texture and structure at the image level.

## 3.1 Method

Given a set of NeRFs denoted by $\{f_{(\phi_n, h_n)}\}_{n=1}^N$, where $N$ denotes the number of object instances in a given object category, we want to learn a prior $\Phi = \{\Phi_S, \Phi_C\}$, where $\Phi_S$ and $\Phi_C$ are the shape and color priors, respectively. Each NeRF, $f_{(\cdot)_n}$, is parameterized by the neural network weights, $\phi_n$, and learnable MRHEs, $h_i$ as proposed in [35]. $f_{(\cdot)_n}$ takes a 3D position, $\mathbf{x}$, and viewing direction, $\theta$, as input and predicts the density conditioned on $\mathbf{x}$ denoted by $\sigma_n^{\{\mathbf{x}\}}$, and color conditioned on $\mathbf{x}$ and $\theta$ denoted by $c_n^{\{\mathbf{x}, \theta\}}$. This is given as,

$$f_{(\phi_n, h_n)}(\mathbf{x}, \theta) = \{\sigma_n^{\{\mathbf{x}\}}, c_n^{\{\mathbf{x}, \theta\}}\}. \tag{1}$$

Our proposed method for learning NeRF priors involves two steps. First, we train a hypernetwork, $M$, to learn a prior over a set of multiview consistent NeRFs of high-quality shape and texture. Second, we employ an image-based denoising network that takes as input an already multiview consistent set of images, rendered from the predicted NeRF, and improves the shape and texture of NeRF to higher quality by finetuning on a set of denoised images. Our architecture is outlined in Figure 3 and we explain each step in detail below.

**Step 1: Hypernetwork for Learning NeRF Prior.** We want to design our hypernetwork, $M$, with trainable parameters, $\Omega$ that can predict NeRF parameters $\{\phi_n, h_n\}$ given a conditioning code $z_n = \{S_n, C_n\}$, where $S_n$ and $C_n$ are the shape and color codes, respectively, for an object instance $n$ belonging to a specific category. Here, $S_n$ and $C_n$ belong to codebooks, $S$ and $C$ that are trained along with $\Omega$ in an auto-decoding fashion.

As shown in Figure 3 (top), ideally we want $M$ to learn a prior $\{\Phi_C, \Phi_S\}$ over $S$ and $C$ such that given a random set of codes, $\{\mathcal{Y}_S \sim \Phi_S, \mathcal{Y}_C \sim \Phi_C\}$, $M$ should be able to generate a valid NeRF with consistent shape and texture for the given category of objects. To achieve this, we train $M$ by assuming the same constraints as are needed to train a NeRF - a set of multiview consistent images $\mathbf{I} = \{\{I_{\theta \in \Theta}\}_n\}_{n=1}^N$ for a set of poses, $\Theta$. In each training step, we start with a random object instance, $n$, and use the corresponding codes $S_n$ and $C_n$ from the codebooks as an input for $M$. Our key insight is that estimating **both** the MRHEs and MLP weights results in a higher quality than other alternatives. $M$ then predicts the NeRF parameters $\{\phi_n, h_n\}$, which is then used to minimize the

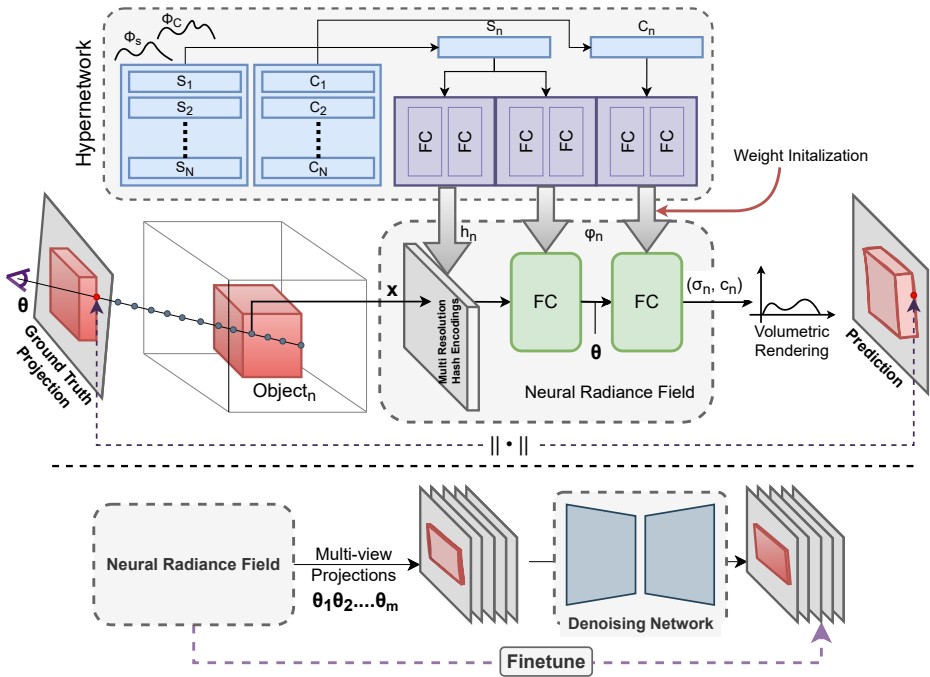

Figure 3: **Architecture Diagram:** HyP-NeRF is trained and inferred in two steps. In the first step **(top)**, our hypernetwork, $M$, is trained to predict the parameters of a NeRF model, $f_n$ corresponding to object instance $n$. At this stage, the NeRF model acts as a set of differentiable layers to compute the volumetric rendering loss, using which $M$ is trained on a set of $N$ objects, thereby learning a prior $\Phi = \{\Phi_S, \Phi_C\}$ over the shape and color codes given by $S$ and $C$, respectively. In the second step **(bottom)**, the quality of the predicted multiview consistent NeRF, $f_n$, is improved using a denoising network trained directly in the image space. To do this, $f_n$ is rendered from multiple known poses to a set of images that are improved to photorealistic quality. $f_n$ is then finetuned on these improved images. Importantly, since $f_n$ is only finetuned and not optimized from scratch, and thus $f_n$ retains the multiview consistency whilst improving in terms of texture and shape quality.
following objective:

$$\mathcal{L}(\Omega, S_n, C_n) = \sum_{\mathbf{r} \in R} ||\mathbf{V}'(\mathbf{r}, \{\sigma_n^{\{x_i^{\mathbf{r}}\}}, c_n^{\{x_i^{\mathbf{r}}, \theta\}}\}_{i=1}^{L}) - \mathbf{V}_n(\mathbf{r})|| \tag{2}$$

$$\{\sigma_n^{\{x_i^{\mathbf{r}}\}}, c_n^{\{x_i^{\mathbf{r}}, \theta\}}\} = f_{(\phi_n, h_n)}(x_i^{\mathbf{r}}, \theta) \quad \text{and} \quad \{\phi_n, h_n\} = M_{\Omega}(S_n, C_n) \tag{3}$$

where $\mathbf{V}'$ denotes the volumetric rendering function as given in [32] eqn. 3 and 5, $\mathbf{r}$ is a ray projected along the camera pose $\theta$, $x_i^{\mathbf{r}} \in \mathbf{x}$ and $L$ denote the number of points sampled along $\mathbf{r}$, and $\mathbf{V}_n$ denote the ground truth value for projection of the $n^{\text{th}}$ object along $\mathbf{r}$.

Note that, in this step, the only trainable parameters are the meta-network weights, $\Omega$, and the codebooks $S$ and $C$. In this setting, the NeRF functions $f_{(\cdot)_n}$ only act as differentiable layers that allow backpropagation through to $M$ enabling it to train with multiview consistency loss attained by the volumetric rendering loss as described in [32]. We use an instantiation of InstantNGP [35] as our function $f_{(\cdot)_n}$ consisting of MRHE and a small MLP.

A general limitation of hypernetworks arises from the fact that the intended output space (i.e. the space of valid MLP weight matrices) is a subset of the actual output space, which is unristricted and can be any 2D matrix. Thus, a hypernetwork trained on loss functions in the weight space can result in unstable training, and might require a lot of training examples to converge. To overcome this issue, we train our hypernetwork end-to-end directly on images, so that it learns the implicit NeRF space along with the category specific prior on it, which simplifies the setting for the hypernetwork and allows for more stable training. As a causal effect of this, HyP-NeRF, when trained on less number of examples, essentially acts as a compressing model.

**Step 2: Denoise and Finetune.** In the first step, $M$ is trained to produce a consistent NeRF with high-fidelity texture and shape. However, we observed that there is room to improve the generated

|  |  | Chairs | | | | Sofa | | | |
|---|---|---|---|---|---|---|---|---|---|
|  |  | PSNR↑ | SSIM↑ | LPIPS↓ | FID↓ | PSNR↑ | SSIM↑ | LPIPS↓ | FID↓ |
| ABO-512 | PixelNeRF [75] | 18.30 | 0.83 | 0.31 | 292.32 | 17.51 | 0.84 | 0.28 | 323.89 |
|  | CodeNeRF [19] | 19.86 | 0.87 | 0.298 | - | 19.56 | 0.87 | 0.290 | - |
|  | HyP-NeRF (Ours) | **24.23** | **0.91** | **0.16** | **68.11** | **23.96** | **0.90** | 0.18 | **120.80** |
|  | w/o Denoise | 23.05 | 0.90 | **0.16** | 102.45 | 23.54 | **0.90** | **0.174** | 121.69 |

Table 1: **Generalization**. Comparison of single-posed-view NeRF generation. Metrics are computed on renderings of resolution $512 \times 512$. HyP-NeRF significantly outperforms PixelNeRF and CodeNeRF on all the metrics in both the datasets.

NeRFs to better capture fine details like uneven textures and edge definition. To tackle this challenge, we augment $M$ using a denoising process that takes $f_{(\cdot)_n}$ and further finetunes it to achieve $f^H_{(\cdot)_n}$.

As shown in Figure 3 (bottom), we render novel views from the multiview consistent NeRF into $m$ different predefined poses given by $\{\theta_1, \theta_2...\theta_m\}$ to produce a set of multiview consistent images $\{\hat{I}_i\}_{i=1}^m$. We then use a pre-trained image-level denoising autoencoder that takes $\{\hat{I}_i\}_{i=1}^m$ as input and produces images of improved quality given as $\{\hat{I}_i^H\}_{i=1}^m$. These improved images are then used to finetune $f_{(\cdot)_n}$ to achieve $f^H_{(\cdot)_n}$. Note that, we do not train the NeRFs from scratch on $\{\hat{I}^H\}$ and only finetune the NeRFs, which ensures fast optimization and simplifies the task of the denoising module that only needs to improve the quality and does not necessarily need to maintain the multiview consistency. While our denoising is image-level, we still obtain multiview consistent NeRFs since we finetune on the NeRF itself (as we also demonstrate through experiments in the supplementary).

For our denoising autoencoder, we use VQVAE2 [42] as the backbone. To train this network, we simply use images projected from the NeRF, predicted by the hypernetwork (lower quality relative to the ground truth) as the input to the VQVAE2 model. We then train VQVAE2 to decode the ground truth by minimizing the L2 loss objective between VQVAE2's output and the ground truth.

## 3.2 HyP-NeRF Inference and Applications

Training over many NeRF instances, $M$ learns a prior $\Phi$ that can be used to generate novel consistent NeRFs. However, $\Phi$ is not a known distribution like Gaussian distributions that can be naively queried by sampling a random point from the underlying distribution. We tackle this in two ways:

**Test Time Optimization**. In this method, given a single-view or multi-view posed image(s), we aim to estimate shape and color codes $\{S_o, C_o\}$ of the NeRF that renders the view(s). To achieve this, we freeze $M$'s parameters and optimize the $\{S_o, C_o\}$ using the objective given in Equation (2).

**Query Network**. We create a query network, $\Delta$, that maps a point from a known distribution to $\Phi$. As CLIP's [41] pretrained semantic space, say **C**, is both text and image aware, we chose **C**, as our known distribution and learn a mapping function $\Delta(z \sim \mathbf{C}) \rightarrow \Phi$. Here, $\Delta$ is an MLP that takes $z$ as input and produces $\mathcal{Y}_z \in \Phi$ as output. To train $\Delta$, we randomly sample one pose from the ground truth multiview images $I_\theta^n \in \{I_{\theta \in \Theta}\}_n$ and compute the semantic embedding $z_\theta^n = \text{CLIP}(I_\theta^n)$ and map it to $\{\bar{S}_n, \bar{C}_n\} \in \Phi$ given as $\{\bar{S}_n, \bar{C}_n\} = \Delta(z_\theta^n)$. We then train our query network by minimizing the following objective:

$$\mathcal{L}_\Delta = \sum_\theta ||\{\bar{S}_n, \bar{C}_n\}, \{S_n, C_n\}||. \tag{4}$$

At the time of inference, given a text or image modality such as a text prompt, single-view unposed (in-the-wild) image, or segmented image, we compute the semantic embedding using CLIP encoder and map it to $\Phi$ using $\Delta$, from which we obtain the shape and color codes as input for the HyP-NeRF.

Note that for N query points in a scene, the forward pass through the hypernetwork (computationally expensive) happens only once per scene. Only the NeRF predicted by the hypernetwork (computationally less expensive) is run for each query point.

## 4 Experiments

We provide evaluations of the prior learned by HyP-NeRF specifically focusing on the quality of the generated NeRFs. We consider three dimensions: (1) **Generalization** (Section 4.1): we validate whether HyP-NeRF can generate novel NeRFs not seen during training by conditioning on only a

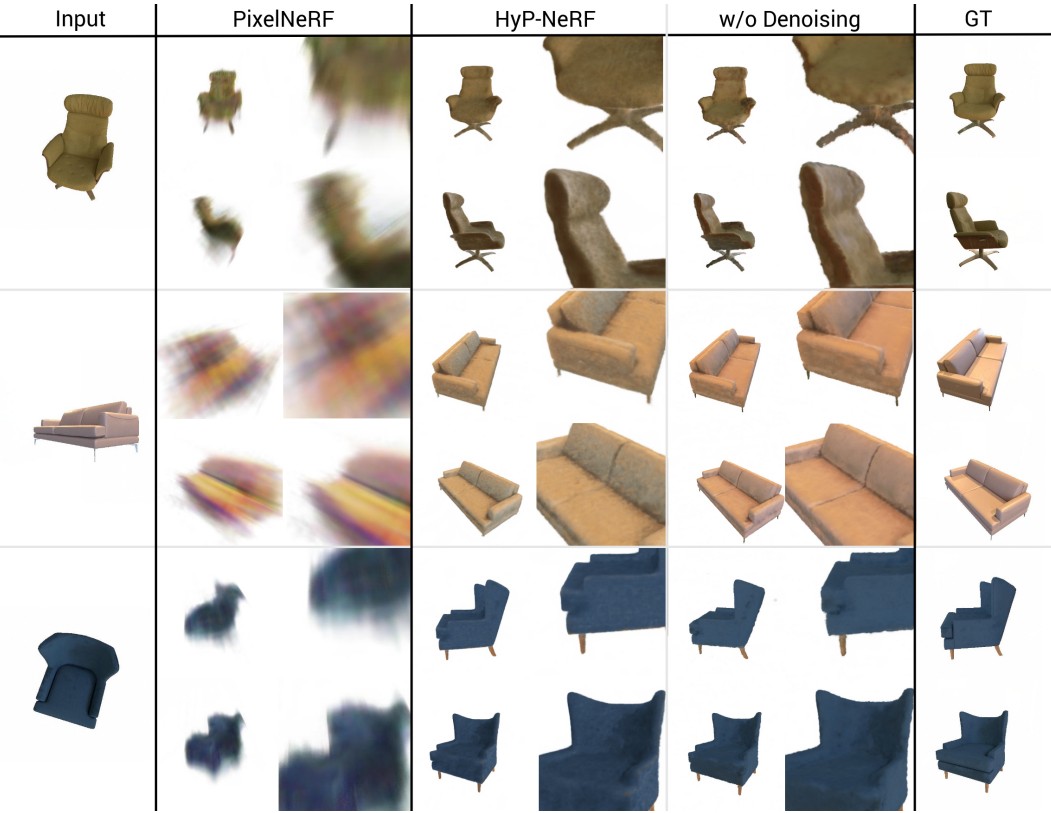

| Input | PixelNeRF | HyP-NeRF | w/o Denoising | GT |

Figure 4: **Qualitative Comparison of Generalization on ABO.** The NeRFs are rendered at a resolution of $512 \times 512$. HyP-NeRF is able to preserve fine details such as the legs, creases, and texture even for novel instances. PixelNeRF fails to preserve details and to model the structure.

| | ABO Chairs | | | | ABO Table | | | | ABO Sofas | | | |
|---|---|---|---|---|---|---|---|---|---|---|---|---|
| | PSNR↑ | SSIM↑ | LPIPS↓ | CD↓ | PSNR↑ | SSIM↑ | LPIPS↓ | CD↓ | PSNR↑ | SSIM↑ | LPIPS↓ | CD↓ |
| [35] | 35.43 | 0.96 | 0.07 | – | 34.07 | 0.95 | 0.07 | – | 33.87 | 0.95 | 0.08 | – |
| Ours | 31.37 | 0.94 | 0.1 | 0.0082 | 29.52 | 0.93 | 0.11 | 0.0033 | 30.32 | 0.94 | 0.11 | 0.0118 |

Table 2: **Compression**. We randomly sample 250 datapoints from our training dataset and compare the NeRFs learned using InstantNGP [35] on the individual instances against HyP-NeRF that learns the entire dataset. Note, we do not employ the denoising module (see Section 3.1) for this evaluation.

single-posed-view of novel NeRF instances. (2) **Compression** (Section 4.2): since HyP-NeRF is trained in an auto-decoding fashion on specific NeRF instances (see Equation (2)), we can evaluate the quality of the NeRFs compressed in this process. (3) **Retrieval** (Section 4.3): as shown in Figure 2, HyP-NeRF's prior enables various downstream applications. We show how to combine our prior with CLIP [41] to retrieve novel NeRFs.

**Datasets and Comparisons**. We primarily compare against two baselines, PixelNeRF [75] and InstantNGP [35] on the Amazon-Berkeley Objects (ABO) [11] dataset. ABO contains diverse and detailed objects rendered at a resolution of $512 \times 512$ which is perfect to showcase the quality of the NeRF generated by HyP-NeRF. Rather than use a computationally expensive model like VisionNeRF (on the SRN [56] dataset) on a resolution of $128 \times 128$, we show our results on $512 \times 512$ and compare with PixelNeRF. Additionally, we compare with the other baselines on SRN at $128 \times 128$ resolution qualitatively in the main paper (Figure 5) and quantitatively in the supplementary. For compression, we directly compare with InstantNGP [35], that proposed MRHE, trained to fit on individual objects instance-by-instance.

**Architectural Details**. We use InstantNGP as $f_{(\cdot)_n}$, with 16 levels, hashtable size of $2^{11}$, feature dimension of 2, and linear interpolation for computing the MRHE; the MLP has a total of 5, 64-dimensional, layers. We observed that a hashtable size $2^{11}$ produces NeRF of high-quality at par with

| Input | PixelNeRF | VisionNeRF | Ours | | Input | PixelNeRF | VisionNeRF | Ours |

Figure 5: **Qualitative Comparison of Generalization on SRN** on the task of single-view inversion (posed in our case) and compare the quality of the views rendered at $128 \times 128$. HyP-NeRF renders NeRFs of similar quality to the PixelNeRF and VisionNeRF baselines.

the a size of $2^{14}$. Hence, we use $2^{11}$ to speed up our training. Our hypernetwork, $M$, consists of 6 MLPs, 1 for predicting the MRHE, and the rest predicts the parameters $\phi$ for each of the MLP layers of $f$. Each of the MLPs are made of 3, 512-dimensional, layers. We perform all of our experiments on NVIDIA RTX 2080Tis.

**Metrics**. To evaluate NeRF quality, we render them at 91 distinct views and compute metrics on the rendered images. Following PixelNeRF, we use PSNR($\uparrow$), SSIM($\uparrow$), and LPIPS($\downarrow$) [78]. Additionally, we compute Fréchet Inception Distance (FID)($\downarrow$) [17] to further test the visual quality. Although these metrics measure the quality of novel-view synthesis, they do not necessarily evaluate the geometry captured by the NeRFs. Therefore, we compute Chamfer's Distance (CD) whenever necessary by extracting a mesh from NeRF densities [30]. Please see the supplementary for additional details.

| ABO Chairs | | ABO Sofa | |
|---|---|---|---|
| Top 1 | Top 3 | Top 1 | Top 3 |
| 98.72% | 99.81% | 91.6% | 95.27% |

Table 3: **Retrieval**. We design a simple query network (see Section 3.2) to retrieve NeRF instances from HyP-NeRF's prior seen at the time of training and achieve almost 100% accuracy.

## 4.1 Generalization

One way to evaluate if HyP-NeRF can render novel NeRF instances of high quality is through unconditional sampling. However, our learned prior $\Phi$ is a non-standard prior (like a Gaussian distribution) and thus random sampling needs carefully designed mapping between such a known prior and $\Phi$. Therefore, we instead rely on a conditional task of single-view novel NeRF generation: given a single arbitrarily-chosen view of a novel object, we generate the corresponding NeRF, $f_{(\cdot)_o}$ through test-time optimization (see Section 3.2). We compare quantitatively with PixelNeRF on ABO at a high resolution of $512 \times 512$ and qualitatively with the rest of the baselines on SRN at $128 \times 128$.

As shown in Table 1, we significantly outperform PixelNeRF on all of the metrics. Further, the qualitative results in Figure 4 clearly shows the difference between the rendering quality of HyP-NeRF against PixelNeRF. Specifically, PixelNeRF fails to learn details, especially for the Sofa category. On the other hand, HyP-NeRF preserves intricate details like the texture, legs, and folds in the objects even at a high resolution. Further, we show our results on the widely used SRN dataset at the resolution of $128 \times 128$ in Figure 5. Here, our quality is comparable with the baselines.

## 4.2 Compression

Unlike InstantNGP, which is trained on a single 3D instance, HyP-NeRF is trained on many NeRF instances which effectively results in the compression of these NeRFs into the latent space (or the codebook). We evaluate this compression capability by computing NeRF quality degradation compared to single-instance-only method, InstantNGP.

We randomly sample 250 instances from the training set and train InstantNGP separately on each of them. These samples are a subset of the training data used in HyP-NeRF's codebook. We show degradation metrics in Table 2. Note that we **do not perform denoising** on the generated NeRFs as we want to only evaluate the compression component of HyP-NeRF in this section. As can be seen in Table 2, there is a significant degradation in terms of PSNR (an average of 11%), but the overall geometry is preserved almost as well as InstantNGP. However, InstantNGP is trained on a single instance, whereas we train on 1000s of NeRF instances (1038, 783, and 517 instances for ABO Chairs, Sofa, and Tables, respectively). This results in a $60\times$ compression gain: for ABO Chairs,

with 1038 training instances, HyP-NeRF needs 163MB to store the model, whereas a single instance of InstantNGP needs on average 8.9MB. Note that we use the same network architecture [1] for HyP-NeRF and InstantNGP making this a fair comparison. Moreover, the storage complexity for InstantNGP-based NeRFs is linear with respect to the number of instances, whereas our degradation in visual quality is sublinear.

## 4.3 Retrieval

A generalizable learned prior has the ability to generate NeRFs based on different input modalities like text, images, segmented and occluded images, random noise, and multi-view images. We now demonstrate additional querying and retrieval capabilities as described in Section 3.2.

This experiment's goal is to retrieve specific NeRF instances that HyP-NeRF has encountered during training from a single-view unposed image of that instance. Section 4.3 presents the number of times we could correctly retrieve from an arbitrary view of seen NeRF instances. We achieve almost 100% accuracy for Chair and Sofa datasets. However, we take this a step further and try to retrieve the closest training instance code corresponding to **unseen views** of seen instances taken from in-the-wild internet images. Figure 6 (top) shows examples from this experiment in which we are able to retrieve a NeRF closely matching the input query. This demonstrates the ease of designing a simple mapping network that can effectively interact with HyP-NeRF's prior.

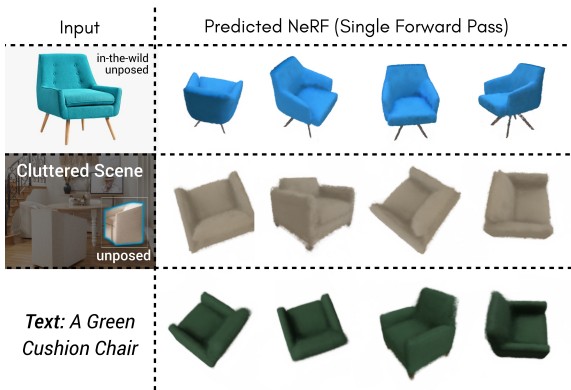

Figure 6: **Qualitative Comparison of Querying (Section 3.2) on HyP-NeRF's prior**. In the top, we use an in-the-wild single-view unposed image to retrieve the closest NeRF HyP-NeRF has seen during training. In middle, we take a cluttered scene, and mask out the object of interest using Segment Anything [22] and in the bottom we use a text prompt as an input to our query network, $\Delta$. We then obtain the latent codes $\{S, C\}$ from $\Delta$, which are used as an input for HyP-NeRF.

Along with retrieving a seen instance, we use the query network to generate novel NeRFs of **unseen instances** as shown in Figure 6 (middle and bottom). In the middle row, we take an image of a cluttered scene, segment it with SAM [22], and pass this as input to the query network, from which we obtain a set of latent codes given as input to HyP-NeRF (see Figure 2). Finally, in the bottom row, we show text-to-NeRF capabilities enabled by HyP-NeRF.

## 4.4 Ablation

Two key designs of HyP-NeRF include incorporating the MRHE and the denoising network. We present the affect of removing these two components in Table 4 and Figure 1 for MRHE and Table 1, Figure 2, and Figure 4 for denoising. In the first ablation, we change the design of our neural network by using a hypernetwork to predict the parameters of a standard nerf with positional encodings [32]. Since we remove the MRHE, we also increase the number of layers in the MLP to match the layers mentioned in [32]. Since there is a significant increase in the view rendering time, we randomly sam-

|  | Chairs | | | |
|---|---|---|---|---|
|  | PSNR↑ | SSIM↑ | LPIPS↓ | CD↓ |
| HyP-NeRF | **29.23** | **0.94** | **0.10** | **0.0075** |
| w/o MRHE | 26.42 | 0.92 | 0.16 | 0.0100 |

Table 4: **Ablation of removing MRHE** on ABO dataset. Due to the significant rendering time of HyP-NeRF w/o MRHE, we sample 70 object instances from the training dataset to compute the metrics at $512 \times 512$ resolution.

ple 70 training examples for evaluation. As seen in Table 4, the quality of the rendered views lags significantly in all the metrics including the CD (measured against NeRFs rendered on InstantNGP individually). This is showcased visually in Figure 1 and the supplementary. Similarly, we find significant differences between the quality of the NeRFs before and after denoising (Table 1, Figure 2, and Figure 4), particularly in the Chair category with more diverse shapes.

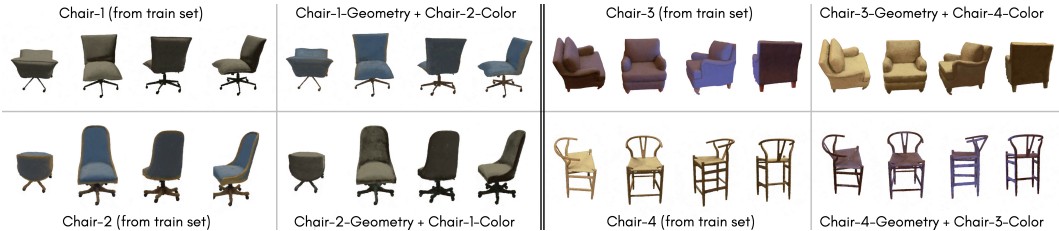

Figure 7: Qualitative results on color and geometry disentanglement: We take two instances from the training set and switch the geometry and color codes to generate novel instances. As can be seen, the geometry and the color are transferred while preserving fine shape details. Even the fine details, like stripes and color-contrast between the chair seats and edges from Chair-2, are accurately transferred to Chair-1 (Chair-2-Geometry + Chair-1-Color). **Zoom in for an improved experience.**

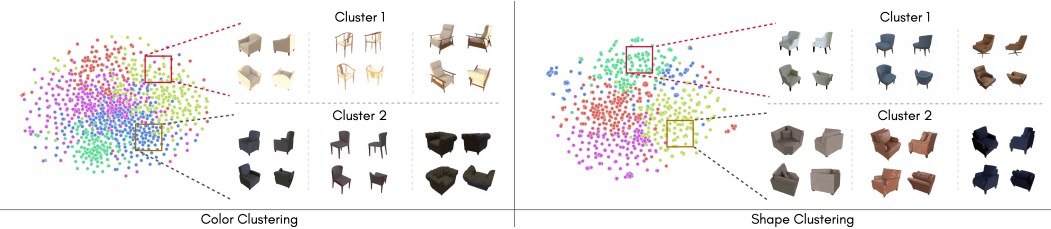

Figure 8: Latent visualization through TSNE plots on shape and color codes (from the codebooks $S$ and $C$). As can be seen, the underlying space forms meaningful clusters as shown through examples randomly sampled from two different clusters.

### 4.5 Color and Shape Disentanglement

We start with two object instances from the train set - *A* and *B* and denote their corresponding shape and geometry codes as $A_s$, $B_s$ and $A_g$, $B_g$. We switch the geometry and shape code and generate two novel NeRFs given by $A_s$, $B_g$ and $B_s$, $A_g$. In Figure 7, we can clearly see the disentanglement: geometry is perfectly preserved, and the color is transferred faithfully across the NeRFs. In Figure 8, we show clusters of color and shape codes using TSNE plots and visualize instances from the clusters.

## 5 Conclusion, Limitation, and Future Work

We propose HyP-NeRF, a learned prior for Neural Radiance Fields (NeRFs). HyP-NeRF uses a hypernetwork to predict instance-specific multi-resolution hash encodings (MRHEs) that significantly improve the visual quality of the predicted NeRFs. To further improve the visual quality, we propose a denoising and finetuning technique that result in an improved NeRF that preserves its original multiview and geometric consistency. Experimental results demonstrate HyP-NeRF's capability to generalize to unseen samples and its effectiveness in compression. With its ability to overcome limitations of existing approaches, such as rendering at high resolution and multiview consistency, HyP-NeRF holds promise for various applications as we demonstrate for single- and multi-view NeRF reconstruction and text-to-NeRF.

**Limitation and Future Work**. One limitation of our work is the need for the pose to be known during test-time optimization (Section 3.2). Although we propose the query network to predict novel NeRFs conditioned on an unposed single view, the result may not exactly match the given view because of the loss of detail in the CLIP embedding. Future work should design a mapping network that can preserve fine details. An iterative pose refinement approach that predicts the pose along with the shape and color codes could also be adopted. A second limitation of our work is the non-standard prior $\Phi$ that was learned by HyP-NeRF which makes unconditional generation challenging. GAN-based generative approaches solve this problem by randomly sampling from a standard distribution (like Gaussian distribution) and adversarially training the network. However, those methods often focus more on image quality than 3D structure. Future work could address this by incorporating latent diffusion models that can map a standard prior to HyP-NeRF's prior.

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
