# OpenReview forum: "HyP-NeRF: Learning Improved NeRF Priors using a HyperNetwork"
_NeurIPS.cc/2023/Conference — NeurIPS 2023 poster_

### Official Review · Reviewer_bLah · 2023-06-13

**Soundness:** 2 fair
**Presentation:** 3 good
**Contribution:** 2 fair
**Rating:** 4
**Confidence:** 4

**Summary:**

In the paper authors propose a novel method for NeRF based model which able to generalize. Authors use hypernetwork paradigm and Multi-Resolution Hash Encodings.

The paper is interesting but has two main problems:
- two-stage training, where the second stage is applied to generate NeRF representation.
- authors do not mention a few important works in relation works and with comparison.

**Strengths:**

The generalization ability of NeRF based model is fundamental.
The paper shows an interesting application of the model.

**Weaknesses:**

## Related works lack a few important works:

1. The generalization ability of NeRF based model is significant. I believe the NeRF generalization models can be divided into two groups. Relatively small models which are similar to NeRF architecture like Voxel Base NeRF, tri-plane NeRF or MultiplaneNeRF, Pix2NeRF, and model which uses GAN, autoencoder, or diffusion model combined with NeRF.
The model is in the second group since we use the network to generate another model like in Points2NeRF or  Hypernerfgan.

2. The related work does not mention the tri-plane NeRF mod model, which is now extremely important. The author mentions paper EG3D [7], but the relation between models should be highlighted.

3. There are a few models which use hypernetworks in a similar fusion:

Points2NeRF: Generating Neural Radiance Fields from 3D point cloud
and
Hypernerfgan: Hypernetwork approach to 3d nerf GAN

4. Consequently, the sentence: “However, unlike us, they do not use a hypernetwork, and use the meta-learning algorithms only for initializing a NeRF, which is further fine-tuned on the multiview images” should be corrected.

## Two-stage training

5. In my opinion, the two-stage training is problematic since the second step can be applied in all models. In the second stage, we tune the model produced by the hypernetwork.

6. In my opinion, all tables authors should give results for models with and without the second stage.

7. We need a pre-train autoencoder in the second stage.

## The model train generalizable NeRF priors

8. It is unclear why we do not force laten to be generative by adding some distance to classical Gaussian prior, similar to VAE.

9. Why do we not use an autoencoder instead of a trainable latent with a decoder?

## Experiments

10 . The experimental section is well organized, but in my opinion, the authors should compare models with GANS, autoencoders, and diffusion-based models in some sense.

11. The model should be compared with Points2NeRF, since the architecture is very similar.

**Questions:**


## Uncler parts of the paper

1. Authors claim: “Our key insight is to use hypernetworks to generate both the network weights and instance-specific MRHEs.” and  “Each NeRF, f(·)n , is parameterized by the neural network weights, ϕn, and learnable MRHEs, hi ...”

So Hypernetwork produces hi or hi are trainable?  Such a sentence is misleading.

2. Authors claim: “Given a set of NeRFs denoted by {f(ϕn,hn)}Nn, where N denotes the number of object instances in a given object category, we want to learn a prior Φ = {ΦS,ΦC},”

It suggests that we need the NERF representation of each object in the training dataset. Such a sentence is misleading.

3. Hypernetwork is denoted by M. Usually, we use H for hypernetwork.

4. “We want to design our hypernetwork, M,with trainable parameters, Ω that can predict NeRF parameters {ϕn, hn} given a conditioning code zn = {Sn,Cn}.

Hypernetworks and conditioning mechanisms are not the same as the author notices in the introduction. I recommend to do not mixing such methods.

5. The sentence is unclear: “Here Sn Cn belong to codebooks, and that are trained within an auto-decoding fashion.”

What does it mean?

6. “However, Φ is not a known distribution like Gaussian distributions that can be naively queried by sampling a random point from the underlying distribution. “

Why do we not add some term to loss to force latent to be Gaussian?

**Limitations:**

 The authors discuss the limitations of the model.

---

> ### Author Rebuttal · Authors · 2023-08-09
>
> We would like to thank the reviewer for their thoughtful comments.
>
> 1. **Related works...**: We would like to thank the reviewer for the categorization. However, there could be several criteria for categorizing different techniques, like based on the downstream tasks the technique enables. Further, we will add the suggested related works in the final version. The sentence "However, unlike us, they do not use a hypernetwork..." is written in the context of Learned Intitialization [59], which is correct. We are happy to provide further clarification on this during the discussion phase.
>
> 2. **Two-stage training pipeline**
>
>     - We don’t fully understand why the reviewer thinks it is problematic.
>     - While it is true that the second stage can be applied to the baselines -- **how to apply the second stage to the baselines is not trivial**.
>     - The expected input in our second stage is a standard NeRF whose parameters can be optimized through a standard volumetric loss on the set of denoised images. However, baselines like PixelNeRF and VisionNeRF work with a modified NeRF that is densely conditioned on pixel-level image features for a particular viewpoint (such as dense CNN-based features). In the fine-tuning step for such baselines, one would need to finetune both the NeRF and dense pixel-level feature parameters -- which is non-trivial.
>     - For a baseline like CodeNeRF, which could potentially be fine-tuned, we show the example of the denoising module in the added rebuttal PDF, Figure 4, right. As shown, the VQVAE2 fails to improve the results due to the amount of noise in the input. On the other hand, for HyPNeRF (Figure 4, left), VQVAE2 improves the images by refining the edges and improving the texture allowing us to perform fine-tuning on these denoised images.
>
> 3. We have added results with and without denoise and finetune in the rebuttal PDF Table 1 and 3.
>
> 4. **Pretrained autoencoder:** The autoencoder (VQVAE2) can be trained easily using images rendered from HyP-NeRF (as the input) and the ground truth NeRF renderings. We do not depend on any additional data. Moreover, the overall training regime is simple to follow. Given this, we are confident that we have proposed a novel technique that can be a valuable contribution to the NeRF community.
>
> 3. **Design choice**: The technique of auto-decoding is inspired from an array of influential works like DeepSDF [37], Scene Representation Networks (SRN) [54], Light Field Networks [53], INR-V [46], CodeNeRF [17], and many more that learn high-quality prior over implicit neural representations (INRs) - in our case a NeRF. Following are additional reasons why auto-decoding is a more suitable technique to learn a prior in our case -
>
>     - Building an encoder for our model without limiting the tasks is not trivial. For example, including an image-based encoder will add constraints of view and scale. pointcloud encoders would add a dependency on pointclouds adding a dependency on 3D supervision. Our current design only needs 2D supervision.
>     - Such a training technique would also make it hard to generalize to diverse inputs (such as text and single-or-multiview images). Further, an encoder would need to output two codes - shape and color - while ensuring the disentanglement between them. In our design, we learn a generic prior over NeRF representation (see rebuttal PDF Figures 1, 2, and 3).
>     - We also encourage the reviewer to refer to [54] and [17] for further clarification.
>     - **Adding a loss term to make it Gaussian**: The hypernetwork aims to learn a set of codes corresponding to a set of NeRFs. In this case, one could add a KL divergence loss to force these codes to follow a Gaussian distribution, but without "random sampling" as in Variational AutoEncoders, the space would be too sparse for us to force it to become a Gaussian distribution. Therefore we allow the hypernetwork to learn a prior by itself following the array of influential methods cited in the paper.
>
> 3. **GANs and diffusion-based methods**: HyP-NeRF significantly differs from GAN-based works aiming to generate multi-view consistent images. We aim to produce a NeRF. Incorporating discriminators, GANs are often limited to lower resolutions, whereas HyP-NeRF thrives at a higher resolution of 512. Diffusion-based NeRFs (like DiffRF [31]) rely on an explicit version of NeRFs called radiance field and are thus also limited in the resolution they can output. We produce NeRFs in implicit space. Both of these works primarily show the task of unconditional sampling. On the contrary, our current baselines - PixelNeRF and CodeNeRF - also aim to learn a prior directly over NeRFs and do not provide comparisons with GAN or diffusion-based models.
>
> 4. **Comparison with Points2NeRF**: Even though Points2NeRF follows a similar architecture as ours, this is also true for many other works like DeepSDF [37], LFNs [53], INR-V [46], and so on; However, like these works, Points2NeRF primarily aims to solve a very different task: converting a pointcloud to NeRFs. This would need us to either significantly modify our own method or the proposed Points2NeRF method to be able to compare both of them.
>
> 5. **Unclear parts of the paper.**
>     1. $h_i$ is predicted by the hypernetwork. $f(.)_n$ consists of the MLP parameters $\phi_n$ and MRHE $h_i$ and the hypernetwork predict both of them.
>     2. We will clarify the sentence: we only need multi-view image supervision.
>     3. Many different notations have been followed for denoting hypernetwork in the previous works, $\Psi_\psi$ in Light Field Networks[53], $d_\omega$ in INR-V [46], and so on.
>     4. As the hypernetwork generates the output NeRF based on a given code, $z_i$ - we call it conditioning. This is the same style of notation used in [53] and [46].
>     5. In the auto-decoding setup, $S_n$ and $C_n$ are trainable.
>     6. Addressed in point 3.

---

> > ### Comment · Reviewer_bLah · 2023-08-16
> > **Thank you for your rebuttal.**
> >
> > Thank you for your answer. All my questions and concerns have been addressed, but I think the authors should correct the paper fundamentally. I stay with my score.

---

> > > ### Author Response · Authors · 2023-08-17
> > > **Author's response to reviewer's comment**
> > >
> > > Thank you for engaging in the discussion. If the rebuttal addressed all your concerns, could you please explain what needs to be fundamentally corrected?

---

> > > > ### Comment · Reviewer_bLah · 2023-08-17
> > > > **Thank you for your rebuttal.**
> > > >
> > > > In my opinion, such an idea exists in the literature. I understand your approche is slice different, but NerIPS is one of the best conferences in the field. Therefore, you should correctly pose your model in the literature and process more detailed comparisons with the existing methods.

---

### Official Review · Reviewer_vtAE · 2023-07-01

**Soundness:** 2 fair
**Presentation:** 3 good
**Contribution:** 3 good
**Rating:** 5
**Confidence:** 4

**Summary:**

In this paper, the authors present HyP-NeRF, a framework based on meta-learning principles, tailored for the learning of category-level NeRF priors. This is achieved through conditioning on latent codes with the assistance of hypernetworks. The NeRF model struggles with generalization across categories because of the high dimensionality of network space. To circumvent this issue, the authors propose a novel approach that estimates the multi-resolution-hash encoding and network’s weights via a hypernetwork conditioned on an instance specific latent vector. Furthermore, to denoise the predicted novel views during the fine-tuning phase, they utilize a VQ-VAE-2 model. The authors report significant performance gain in comparison with existing baselines. In addition, the authors showcase HyP-NeRF’s ability in downstream tasks such as single-view novel view generation, text-to-NeRF, and inpainting from cluttered scenes.

**Strengths:**

1.	The authors propose a novel idea for conditioning the network weight and multi-resolution hash encoding via hypernetworks. This approach interestingly facilitates the sharing of prior knowledge within a category level among NeRF instances. Rather than optimizing a fixed set of parameters, the authors present an approach to learn a weight distribution and condition it directly on the instance-level latent code.
2.	By providing a robust category-level prior, HyP-NeRF enables the training of a single model for objects within the same category. This approach effectively eliminates the tendency of overfitting to a single scene as vanilla NeRF and its variants.
3.	Through the conditioning with the latent embedding, HyP-NeRF demonstrates the capacity to cooperate with other models for various downstream tasks. These applications range from text-to-NeRF to novel view synthesis in cluttered scenes.


**Weaknesses:**

1.	My main concern is hypernetwork’s ability to disentangle geometry and color using the latent code Sn and Cn. Taking the single-view reconstruction case as an example (Table 1), this problem is under-constrained as it requires 3D inductive biases learned from a large set of scenes like the target scene. This is apparent in methods such as Pixel-NeRF and its variants, which rely on 2D image features to generalize to unseen scenes after substantial training. However, HyP-NeRF’s approach to resolve the geometric ambiguity through the denoising the 2D outputs does not seem logically coherent. Additionally, it would have been beneficial for the authors to provide evidence showing category geometry for {Sn, Cn} in Section 3.1, and a demonstration of color clustering for scenes with similar colors. Moreover, cases where HyP-NeRF appears to overlook the shape or color of an object (as seen in Figure 6 row 1 and the supplementary video 3:52, 4:06, 4:31) raise further concerns about the ability of latent codes Sn and Cn to capture geometry and color.
2.	The scope of the baseline and dataset appears insufficient to conclusively support the authors’ claim, which echoes my previous comment that HyP-NeRF only benchmarked against Pixel-NeRF under the ABO dataset. Comparative results with more baselines and datasets, such as VisionNeRF[1] and NeRFDiff[2], would have added credibility to their claims. This becomes particularly relevant given that in the supplementary material, HyP-NeRF does not outperform the baselines on the SRN dataset (e.g., PSNR of 21.02 versus 24.48 with Vision-NeRF). I do notice that authors claim that for SRN they did not apply denoising finetuning and SRN has different poses, but it would be hard to judge HyP-NeRF’s performance under different settings. Therefore, I would like to see HyP-NeRF’s performance on ShapeNet with other baselines under the same setting to validate its performance.
3.	As the authors introduce hypernetwork, which adds extra computation on top of the original NeRF. I would like to see computation cost in terms of (# MLP parameters) and (# of FLOPS) in comparison with the baselines. This is particularly relevant considering that HyP-NeRF requires test time optimization. Additionally, how does the latent codebook size (Sn and Cn) impact the performance of the network?
4.	Readability issues, for example:
ln 170 (3) should end with comma “,”.
ln 171 uses “eqn.” to denote the equation, but later in ln 202 uses “Equation”.


[1] Vision Transformer for NeRF-Based View synthesis from a Single Input Image https://arxiv.org/pdf/2207.05736.pdf

[2] NeRFDiff: Single-image View synthesis with NeRF-guided distillation from 3D-aware diffusion
https://arxiv.org/pdf/2302.10109.pdf



**Questions:**

Also listed in the weakness section for detailed reasons.
1.	Please run more experiments with different baselines, for example, Vision-NeRF, NeRFDiff, and additional dataset such as ShapeNet to validate HyP-NeRF’s performance.
2.	Conduct ablation studies to explore if Sn and Cn can capture object shape and color, such as similar Sn (same category) or clustered Cn (if they are within the same category and similar color).
3.	Another important aspect is computation budgets and rendering speed. Given that HyP-NeRF requires test time optimization, which Pixel-NeRF does not, a comparison between the two models’ computational requirements and rendering speed would be highly informative.


**Limitations:**

Yes, the authors have included limitation. Social impact does not apply.

---

> ### Author Rebuttal · Authors · 2023-08-09
>
> We sincerely appreciate the reviewer's thorough analysis of our work and for raising their questions and concerns. In the global and local rebuttal, we have addressed the concerns and questions and provided additional comparisons and visualizations (rebuttal PDF).
>
> 1. To demonstrate **shape and color disentanglement**, we add three qualitative results in the rebuttal PDF:
>     - Figure 1,
>         - We start with two object instances from the train set - $A$ and $B$ and denote their corresponding shape and geometry code as $A_s$, $B_s$, and $A_g$, $B_g$.
>         - Next, we switch the geometry and shape code and generate two novel NeRFs given by $\\{A_s, B_g\\}$ and $\\{B_s, A_g\\}$.
>         - Here, we can clearly see the disentanglement: geometry is perfectly preserved, and the color is transferred faithfully across the NeRFs.
>     - Figure 2: We fix a geometry code and interpolate the color codes. As shown, the geometry is perfectly preserved while the color smoothly transitions.
>     - In Figure 3, we cluster color and shape codes using TSNE plots and visualize instances from the clusters. As shown, each cluster represents a similar color or shape.
>
>     **HyP-NeRF appears to overlook the shape or color:** We propose two methods to query our learned prior - (1) test-time optimization (TTO) and (2) mapping network (MapN).
>
>       - TTO uses pixel-wise difference between the generated NeRF renderings and ground truth view.
>       - In the MapN, we encode the given single-view image (or text) using CLIP [39] encoder to obtain a feature vector. This feature vector is mapped to the hyper network's learned prior (separate mappings for shape and color codes), allowing the hypernetwork to generate the NeRF in a single forward pass.
>     - In MapN, we lose out on the **low-level details like fine shape and color details** through CLIP's encoding process. Therefore, the resultant generated NeRF does not exactly match the given input, as mentioned in the limitation section on Page 9, line 349.
>         - This indicates a limitation of the mapping technique **not** the prior learned by the hypernetwork as evident through TTO that operates within a generator's prior space. For this reason, we can obtain results on single (or multi)-view images, **even on views with severe occlusion** (video timestamps 5.39, 5.57, 3.51) that match the given input exactly.
>
> 2. **The scope of the baseline and dataset appears ...**
>
>     - R3 suggests two baselines - NeRFDiff and VisionNeRF.
>         - NeRFDiff has been recently accepted to ICML 2023 (after our submission to NeurIPS), was submitted to arxiv very close to the  NeurIPS submission date, and hasn't released its codebase or dataset split on ABO yet. We will add NeRFDiff as a concurrent work in our related work section.
>         - VisionNeRF requires an exorbitant amount of compute, quoting from their paper - “16 NVIDIA A100 GPUs, where the training converges at 500K iterations” for training at a resolution of 128 x 128 on SRN. On the other hand, we train HyP-NeRF on a single NVIDIA 2080Ti RTX GPU. **HyP-NeRF specifically thrives and distinguishes itself at a higher resolution of 512 made of data with high-fidelity textures and shapes** - a resolution of 512 would need much more compute than 128  to train VisionNeRF.  In the supplementary, we have presented comparisons against CodeNeRF, FE-NVS, and VisionNeRF on SRN.
>     - To further add credibility to HyP-NeRF, **we compare it to another popular baseline - CodeNeRF - that employs a similar latent conditioning technique by modifying a NeRF to be a conditional NeRF, on ABO in the rebuttal PDF Table 4.** HyP-NeRF significantly outperforms CodeNeRF at both resolutions.
>
> 3. **Baselines under the same setting**: We present our results on SRN with "denoise and finetune" in the rebuttal PDF, Table 1. We outperform the baselines on the Cars subset. Although we do not outperform on the Chair subset, the following points should be noted -
>     - We chose single-view NeRF as one of the several tasks we can perform to showcase generalization to novel NeRFs. The existing baselines are specifically trained for the task of single-view NeRF generation and modify the NeRF function by conditioning the NeRFs on additional features (like CNN features in PixelNeRF and VisionNeRF). Our NeRF is a standard NeRF that expects only the viewing direction and 3D point location as input. Thus, unlike the baselines, our NeRFs can be directly adopted in any downstream tasks that expect a standard NeRF as input.
>     - We incorporate test-time-optimization, which is known to fail at views that do not provide sufficient context. To further provide evidence of this, we compare with PixelNeRF (without the denoise and fine-tune step for a fair comparison) on **sparse-view NeRF generation ranging from a single view to 5 different views in the rebuttal PDF Table 3**. Our results significantly jump with two views and improve further as the views increase. This is also shown qualitatively in the supplementary paper, Figure 2.
>     - Finally, HyP-NeRF showcases many diverse downstream tasks (including many novel tasks not shown before for NeRFs) through many examples in the submitted papers and video ranging from compression, text-to-NeRF generation, generating NeRF from occluded and cluttered images **scraped directly from the internet without any preprocessing**, and so on.
>
> 5. **how does the latent codebook size (Sn and Cn) ...**: The codebook size is equivalent to the number of training instances $\times$ 512 where each code corresponds to one training instance. Therefore, reducing the codebook size will result in lower generalization on unseen datapoints as it might have seen less number of examples during training. We would be happy to discuss and clarify this further in the discussion phase.
>
> 6. We will replace eqn. with Equation. in the final version of the paper.
>
> Rest of the concerns are addressed in the global rebuttal.

---

> > ### Author Response · Authors · 2023-08-12
> > **Minor correction**
> >
> > We would like to add a small correction to the rebuttal. In point 1, we have mentioned $A_g$ and $A_s$ to be geometry and shape codes respectively, wherein they should be $A_g$ and $A_c$ denoting the geometry (i.e. shape) and the **color** of the object instead. Same goes for instance $B$. This is also mentioned in the added rebuttal PDF.

---

> > ### Comment · Reviewer_vtAE · 2023-08-16
> >
> > Thank you for providing clarifications on your comments. I understand that some of the concerns arise directly from the intrinsic characteristics of the hypernetwork. It is evident that methods integrating hypernetworks into their pipeline would exhibit certain inherent issues. Based on this understanding, I've adjusted my rating to a borderline accept. Here are the issues I'd like to highlight:
> >
> > Although Table 2 demonstrates a reduction in FLOPs during inference, the inference time when considering the full pipeline is notably longer compared to the baselines. Specifically, it amounts to a sum of 312 seconds and 2 minutes.
> >
> > The authors have mentioned that denoising and fine-tuning do not have a significant impact. The global response suggests a "marginal difference". However, when looking at Table 1 and Table 3, I observed a noticeable increase of 9% in the Cars dataset in terms of PSNR when comparing results with and without denoising.

---

> > > ### Author Response · Authors · 2023-08-17
> > > **Author's response to reviewer's comment**
> > >
> > > We are glad that we could address the raised concerns and are delighted to notice the change in rating.
> > >
> > >
> > > Regarding the mentioned issues,
> > > 1. Apologies for the confusion. Here $m$ denotes the number of poses needed to render for the Denoise and Finetune step (main paper, line 183). In this step, we render the NeRF from $m$ views that are denoised and further used for finetuning. Despite the time taken by this step, rendering a full NeRF ($\ge$ 120 views) would be much faster for HyP-NeRF. The exact breakdown is given below -
> > >
> > >
> > >                                     PixelNeRF    CodeNeRF    HyP-NeRF
> > > 	   TTO		 	       -	   305s        165s
> > > 	   Single-view rendering        47.8s          8s          2s
> > > 	   Denoise & Finetune             -             -         319s
> > > 	   (when m = 91)
> > > 	   Total time to render         5736s         1265s       724s
> > >        NeRF from 120 views
> > >
> > >     The total time to render NeRF from 120 views is computed as: TTO $+$ 120 $\times$ time taken for single-view rendering $+$ Denoise & Finetune (for HyP-NeRF). Denoise & Finetune - the denoising step takes 182s (which includes rendering and denoising the 91 views), and finetune step takes 137s, which adds up to 319s.
> > >
> > > 2. In general, we have observed that denoising results in only a marginal difference (thereby retaining the original consistency). However, as you rightly pointed out, the overall improvement (after fine-tuning) is more pronounced, especially on the SRN car dataset. We think this is primarily due to the property of the dataset - the category has less diversity in terms of geometry as compared to chairs or ABO. We would also point out the difference in the final version of the paper.

---

### Official Review · Reviewer_bkzC · 2023-07-02

**Soundness:** 3 good
**Presentation:** 3 good
**Contribution:** 3 good
**Rating:** 5
**Confidence:** 4

**Summary:**

The paper proposes HyP-NeRF, a method using hypernetworks to learn generalizable category-level priors, addressing the limitations of existing work on generalization.  Specifically, the proposed hypernetwork-based method predicts the parameters of NeRF and multi-resolution hash encodings, and further incorporates a denoise and finetune strategy to improve quality while retaining multi-view consistency. Qualitative comparisons and evaluations on three tasks (generalization, compression, and retrieval) show that HyP-NeRF achieves state-of-the-art results.


**Strengths:**

1.	It is an interesting idea to take a hypernetwork to learn the category prior for NeRF, attaching the capacity of generalization to NeRF.

2.	The fine-tuning procedure with a denoising network can improve the texture quality while retaining consistency.

3.	Hyp-NeRF can extend to other downstream tasks, like the single image to 3D and text to 3D.

4.	The paper is well-organized and easy to follow.

5.	The experiments look convincing. It clearly supports the major contribution of the paper that a class of objects can be compressed in a unified network with the help of hypernetwork.


**Weaknesses:**

1.	There are not enough details on why fine-tuning procedures can retain consistency and avoid blurry results. Previous works, like instruct-NeRF2NeRF and StylizedNeRF, claim that fine-tuning with inconsistent images leads to blurry results.

2.	The proposed method focuses on category-specific generalization similar to NeRF-based GAN. They both learn the prior of a class of objects and take NeRF as the 3D representation. Therefore, it is better to conduct an experiment comparing Hyp-NeRF and one of the 3D-aware GANs.

3.	The examples in Figure 4 show low-quality results. Although the examples are simple, the result is not clear, with some noises in the appearance.

4.	It lacks more quantitative comparisons with other methods like Table.4.


**Questions:**

1.	The denoised images seem to be 3D inconsistent. I wonder why the fine-tuning process in the second step can retain the multi-view consistency and bypass the blurry results which are common when fine-tuning NeRF with inconsistent images. Lines 188-191 are not clear.

2.	What about efficiency, i.e., Inference time and training time?


**Limitations:**

1.	The results are low-quality with blurry details and noises. Besides, the examples in the experiments are too simple.

2.	The task is similar to NeRF-based generative models, like pi-GAN and EG3D. The paper lacks a discussion on the comparison between Hyp-NeRF and NeRF-based GAN.

---

> ### Author Rebuttal · Authors · 2023-08-09
>
> We would like to appreciate the reviewer's efforts in studying our work and for their positive assessment of our contributions. Below, we elaborate on the questions raised, and provide additional comparisons to address all the concerns.
>
> 1. **There are not enough details on why fine-tuning procedures can retain consistency and avoid blurry results. Previous works, like instruct-NeRF2NeRF and StylizedNeRF, claim that fine-tuning with inconsistent images leads to blurry results.**
>
>     This is addressed in the global rebuttal
>
> 2. **The proposed method focuses on category-specific generalization similar to NeRF-based GAN. They both learn the prior of a class of objects and take NeRF as the 3D representation. Therefore, it is better to conduct an experiment comparing Hyp-NeRF and one of the 3D-aware GANs.**
>
>     - HyP-NeRF is primarily different in the manner that our main focus is on generating NeRFs, whereas the GANs aim to generate multi-view consistent images of high quality. For example, EG3D first generates an output in a lower resolution of $128 \times 128$ due to a computationally expensive design. They then use a superresolution network to super-resolve the images to $512 \times 512$. Further, these images are passed onto a discriminator, which guides multiview consistency for each image. In this case, the output is an image.  In contrast to EG3D, HyP-NeRF generates a NeRF trained through the volumetric rendering loss, and the resultant output (pre & post denoise and finetune) is a NeRF - that can be adopted in any downstream task.
>
>     - Moreover, compute requirements increase drastically for an architecture involving discriminators.  For example, an entire image should be rendered to be passed on to the discriminator in each forward pass. This would need one to sample number of rays equivalent to image resolution to render an entire image from the NeRF representation. On the other hand, we only need to sample a handful of rays that can fit in the memory in each training iteration and therefore are agnostic of the resolution enabling us to scale up to arbitrary resolutions.
>
>    - Lastly, we are primarily different in the task we aim to show. GAN-based methods heavily focus on unconditional sampling, whereas unconditional sampling is not trivial in our case. Instead, we evaluate the task of single-view NeRF generation with works that closely resemble our works, like CodeNeRF and PixelNeRF, that aim to learn a prior over many NeRF instances.
>
> 3. **The examples in Figure 4 show low-quality results. Although the examples are simple, the result is not clear, with some noises in the appearance.**
>
>     - We would like to point out that Figure 4 has two of HyP-NeRF - (1) one with denoise and fine-tune (2) without denoise and finetune as a part of the ablation. Our final result is the 3rd column. The 4th column presents the ablation results, which are slightly inferior in quality compared to the 3rd. This contrast is also showcased in the video timestamp 2.45 to 3.27.
>     - While we understand quality is subjective, it is important to note that we showcase results on a resolution of 512 while much of the existing works in learning a prior over NeRFs remain in the resolution of 128. 3D GANs like EG3D showcase results in 512, but it is important to note that the method employs superresolution on the outputs of the NeRF renderings (which is originally at 128) that result in multi-view inconsistencies. On the other hand, we directly render at 512 at high quality.
>     - We would be happy to provide further clarification if R2 can point out the exact instances that do not feel up to par in the author-reviewer discussion phase.
>
> 4. **It lacks more quantitative comparisons with other methods like Table.4.**
>
>     - Table 4 is an ablation table where we remove a part of our network design and compare it with the full network to demonstrate the removed part's importance. Therefore, in this table, we have not compared with the baselines. Table 3 is a retrieval experiment using the mapping network to retrieve results from our codebook. Since we are retrieving results from our own codebook, there is no baseline. For the other tables, we have compared with the respective baselines. We would be happy to provide further clarification on this point during the author-reviewer discussion phase.
>     - We have additionally added comparisons with CodeNeRF on ABO (128 and 512 resolutions) in the rebuttal PDF.
>
> 5. **What about efficiency, i.e., Inference time and training time?**
>
>     This is addressed in the global rebuttal.

---

> > ### Comment · Reviewer_bkzC · 2023-08-13
> > **Comment**
> >
> > Thank you for the authors' efforts. Regarding my initial reviews and the author's rebuttal, I would like to provide the following revised response:
> >
> > A1: Thank you for the explanation. I have noticed a minor difference between the results before and after denoising, which leads to a slight inconsistency. This explicitly addresses my concern.
> >
> > A2: Thank you for the comparison. I clearly understand the distinction between the 3D-aware GANs, such as EG3D-like models, and your proposed Hyper-NeRF. However, in my opinion, aside from the EG3D-like models, pure NeRF-based GANs like pi-GAN and subsequent GRAF-HD without super-resolution share intrinsic similarities with Hyper-NeRF. In both cases, a conditioned NeRF is generated by sampling from the learned distribution. Therefore, I still believe it would be beneficial to compare Hyper-NeRF with one of these 3D-aware GANs.
> >
> > A3: I hold the view that high resolution does not equate to high quality. However, I have reconsidered and now acknowledge that the results of Hyper-NeRF can indeed be compared to the recent state-of-the-art methods, although the qualitative results with denoising may not be entirely satisfactory.
> >
> > A4: Thank you for conducting the additional experiments.
> >
> > A5: As indicated in Table 2 of the PDF, Hyper-NeRF achieves comparable inference time after the warm-up phase with the two baselines. However, it's important to note that the warm-up time does impact the overall inference time. With fewer views to render, the difference in performance becomes more pronounced.

---

> > > ### Author Response · Authors · 2023-08-17
> > > **Author's comments to reviewer's response**
> > >
> > > Thank you for your response and for raising further concerns,
> > >
> > >
> > > **A2:** Even though there are dissimilarities, as noted in our rebuttal (based on compute requirements, datasets, and the tasks), we do agree with your point that HyP-NeRF does share fundamental similarities with 3D-aware GANs like pi-GAN and GRAF, as they generate conditioned NeRFs by sampling from learned distributions.
> > >
> > >
> > > Based on your suggestions and our analysis, we have considered the following works in the discussion - GRAF, pi-GAN, GRAM (Deng et. al.), and EpiGRAF (Ivan et. al.). We would, however, like to underline that the difference in compute requirements between GAN-based methods and HyP-NeRF that make the comparison non-trivial --
> > >
> > >   - **Compute Requirements:** Adding to what we mentioned in the rebuttal regarding expensive compute for 3D-aware GANs, EpiGRAF's Table 1 outlines the following -
> > >     - pi-GAN and GRAM go out of memory (OOM) when training on a resolution of 512, even on an NVIDIA V100.
> > >     - EpiGRAF takes 24 V100 GPU days to train on 512 resolution (FFHQ dataset).
> > >
> > >     **Notably, HyP-NeRF is trained on a single NVIDIA RTX 2080 Ti GPU for a period of just 3 days on 512 resolution.**
> > >
> > >   **Comparisons:** We compare with GRAF on the ABO-Chair dataset at a resolution of 512 for the task of single-view NeRF generation through TTO.  GRAF obtains a PSNR and SSIM of $15.87$ and $0.83$, respectively, whereas HyP-NeRF obtains a PSNR and SSIM of $24.23$ and $0.91$, respectively. Furthermore, TTO is performed for ~15 minutes on GRAF on a single instance for the reported results, whereas HyP-NeRF needs only 165 seconds (or 2.75 minutes).
> > >
> > >
> > > **A3:** We do agree with your point of view that high resolution does not equate to high quality. GAN-based methods tend to have higher quality at the cost of high compute, whereas, HyP-NeRF achieves comparable quality at much lower compute.
> > >
> > >
> > > **A5:** The reported inference time (in the rebuttal) denoted the time taken to run the end-to-end pipeline, including the TTO. Specifically, the following is the breakdown
> > >
> > >
> > >                                 PixelNeRF    CodeNeRF    HyP-NeRF
> > > 	TTO		 	       -	   305s        165s
> > > 	Single-view rendering        47.8s          8s          2s
> > > 	Denoise & Finetune             -             -         319s
> > > 	(when m = 91)
> > > 	Total time to render         5736s         1265s       724s
> > >     NeRF from 120 views
> > >
> > > The total time to render NeRF from 120 views is computed as: TTO $+$ 120 $\times$ time taken for single-view rendering $+$ Denoise & Finetune (for HyP-NeRF). *Denoise & Finetune* - the denoising step takes 182s (which includes rendering and denoising the 91 views), and finetune step takes 137s, which adds up to 319s.
> > >
> > > **Our rendering takes just 2 seconds for a single image of 512 resolution** (4$\times$ and 22$\times$ faster than CodeNeRF and PixelNeRF, resp.). This is primarily because our computation is efficiently split between the hypernetwork (that runs only once per instance) and the predicted NeRF, and rendering a view only relies on the latter.

---

> > > > ### Comment · Reviewer_bkzC · 2023-08-18
> > > >
> > > > Thanks for your response. My questions are addressed, and I will stay to my original rating.

---

### Official Review · Reviewer_Muw9 · 2023-07-07

**Soundness:** 3 good
**Presentation:** 3 good
**Contribution:** 3 good
**Rating:** 7
**Confidence:** 4

**Summary:**

This paper proposes a way to encode a categorical prior for NeRFs. The method train an auto-decoder, which optimize together both the latent codes (one for every instance in the dataset) and the decoder parameters. The decoder will take the latent code and predict parameters for an instance-NGP backbone supported NeRF model. One key technical contribution is that the hyper-network predicts not only the parameters for the instant-NGP's MLP, but also the Multi-resolution hash encoding. This allows the predicted NeRF to get higher quality. The paper demonstrates this prior can be used for a variety of tasks.

**Strengths:**

- This method is able to generate relatively high-quality NeRF thanks to the design of hyper-network that predict parameters for both the MRHE and the MLP. This design did circumvent the limitation of storing NeRF at a voxel.
- The paper demonstrated that the learned prior is capable of various tasks. This demonstrate the versatility of this learned prior.
- Novelty. To the best of my knowledge, the solution to build an auto-decoder solution for hyper-network that predicts both the MLP parameters and the MRHE is novel.

**Weaknesses:**

- Require separate pipeline for generation and desnoising. Since the denoising and fine-tuning pipeline is an optimization procedure, it's not clear to me whether the pipeline is able to enforce satisfication of the conditioning input. For example, if we use single image as an input, and would like to obtain a NeRF where certain pose is that image. After test-time optimization using the NeRF loss of the single image, this NeRF will be pipe into the denoise pipeline, which is another optimization. How does this second stage of the pipeline will predict something that's aware of the conditional input signal?
- Hyper-network number of parameters. It occurs to me that the hyper-network can take a lot of parameters in order to predict the parameters for both the MRHE and the MLP. There are also other issue of the hyper-network, such as the output does not necessarily respect the permutation invariance of the MLP parameters space, which is especially the case when the hyper-network structure is only an MLP. Training such a large amount of parameters can require a lot of computing or data.
- It's not clear how to achieve unconditional generation using this pipeline. Since the pipeline, after training, will only provide a codebook and a hyper-network, without a way to sample the distribution of the codebook. This suggests that in order to make this pipeline useful for generating a NeRF that satisfies the data distribution, it requires additional handling of the pipeline such as learning variational auto-decoder.

**Questions:**

See weakness section.

**Limitations:**

The paper has a good discussion of potential limitation. Additional limitation I forsee is already listed in the weakness section.

---

> ### Author Rebuttal · Authors · 2023-08-09
>
> We are delighted to receive such a positive assessment of our work and are grateful for it. Below, we provide the answers to the questions that were raised.
>
> 1. **Require separate pipeline for generation ... this second stage of the pipeline will predict something that's aware of the conditional input signal?**
>
>     This is addressed in the global rebuttal.
>
> 2. **There are also other issue of the hyper-network, such as the output does not necessarily … structure is only an MLP.**
>
>     Unfortunately, we do not understand this part clearly, but would to love to discuss this during the discussion phase.
>
> 3. **Hyper-network number of parameters. It occurs to me that the hyper-network can take a lot of parameters in order to predict the parameters for both the MRHE and the MLP….Training such a large amount of parameters can require a lot of computing or data.**
>
>     This is addressed in the global rebuttal
>
> 4. **It's not clear how to achieve unconditional generation using this pipeline ... pipeline such as learning variational auto-decoder.**
>
>     - As mentioned on page 6, line 219, the hypernetwork is trained in an auto-decoding fashion and consequently learns a non-standard prior. This training strategy of auto-decoding has been well-adopted across an array of influential works that aim to predict an implicit function, such as DeepSDF [37], Scene Representation Networks (SRN) [54], Light Field Networks [53], INR-V [46], and many more. Therefore, as you rightly mentioned, it is not trivial to perform unconditional sampling through our learned prior, which is also listed in the limitation section of our work (page 9 line 346), and we would encourage future works to pursue this direction.
>
>     - One major aspect of unconditional sampling is to showcase generalization. In our case, we test HyP-NeRF's ability to generalize based on a well-adopted conditional task of NeRF generation from a single-view image. Further, we showcase many diverse downstream applications that can be enabled through HyP-NeRF through many examples in the paper and video, including compression, text-to-NeRF generation, generating NeRF from occluded and cluttered images **scraped directly from the internet without any preprocessing** (except the segmentation masks obtained from SAM [20]), and so on.

---

> > ### Comment · Reviewer_Muw9 · 2023-08-12
> > **Discussion about hyper-network**
> >
> > Sorry for not being clear on the hyper-network issues. here is a simple example:
> >
> > Imagine an MLP with 1-D input, one 2D hidden layer, and finally a 1D output and there is no bias. This MLP can be written as $f(x; w_1, w_2) = w_2^T ReLU(x w_1)$, where $w_1, w_2\in R^2$. To use a MLP based hyper-network to predict both $w_1$ and $w_2$, the output of the hyper MLP will be 4 dimensions.
> >
> > However, such structure of the hyper-MLP is not aware of the fact that if we swap the first and second dimension of $w_1$ and $w_2$, the output of the MLP will be the same. This means that the hyper-MLP is predicting a larger space where structure exists, and a common remedy can come from sufficient training with more data.
> >
> > This can be a limitation of the paper if the application doesn’t come with efficient data.

---

> > > ### Author Response · Authors · 2023-08-12
> > > **Clarification about hyper-network**
> > >
> > > We thank you for this clarification. Although, as you rightly mentioned, it could be a limitation of the hypernetworks in general, but in our setting, the training happens in an end-to-end fashion through the volumetric rendering loss. This results in the hypernetwork learning a common permutation of NeRF weights across all the instances, simplifying the task for the hypernetwork. Therefore, it can learn on any number of instances, including a single, two, three, or more instances. However, with such a low number of datapoints, the hypernetwork's prior is too sparse and does not have enough diversity to generalize to novel NeRFs. Therefore, we need to train it on a sufficient number of datapoints (as in the ABO / SRN datasets). We hope this clarifies your question, and we would be happy to discuss this further. We will also add this clarification in the main paper.

---

> > > > ### Comment · Reviewer_Muw9 · 2023-08-16
> > > > **Thanks for the replies!**
> > > >
> > > > Thanks for the replies. I would appreciate in the revision some related discussion of why this kind of hyper-network can work in the application scenarios of interests (like you mention, there is sufficient data points in the dataset).
> > > >
> > > > The rebuttal addresses my concerns. I am intending to keep the accepting rating.

---

> > > > > ### Author Response · Authors · 2023-08-17
> > > > > **Thank you note from the authors**
> > > > >
> > > > > We appreciate your feedback and are glad the rebuttal has addressed your concerns. Thank you for maintaining the acceptance rating. Your suggestions and insights are valuable in enhancing the quality of our work! As mentioned by the reviewer, we will certainly include a discussion on why hyper-network can work in these application scenarios.

---

### Author Rebuttal · Authors · 2023-08-09

We thank the reviewers for their feedback and positive assessment of our contributions. We address major factual errors and broad reviewer concerns in this global response and specific reviewer concerns/questions in the local response. We are confident that we will have addressed all their concerns with this rebuttal and additional visualizations and experiments. We would be happy to clarify any more questions during the author-reviewer discussion phase.

**We refer Muw9 as R1, bkzC as R2, vtAE as R3, bLah as R4**

**Clearly, the reviewers appreciate several aspects of the paper**. R1, R2, and R3 appreciate that HyP-NeRF enables **several downstream tasks** (R1: “demonstrates the versatility of this learned prior”, R2: “can extend to other downstream tasks”, and R3: “cooperate with other models for various downstream tasks”). R1 & R3 find the **idea novel** (R1: “both the MLP parameters and the MRHE is novel” and R3: “authors propose a novel idea for…”). R2 also likes the idea of **denoising and fine-tuning** (“denoising networks can improve the texture quality while retaining consistency”). R3 & R4 appreciate our **experimental setup** (R3: “The experiments look convincing. It clearly supports the major contribution of the paper”, R4: “experimental section is well organized”). R1 finds the NeRFs to be **high quality** (“generate relatively high-quality NeRF thanks to…”).

1. **Factual Errors & Opinions:** We first want to address R4’s review, which, unfortunately, has several factual errors and opinions unsupported by facts. For example:

      - One of the major weaknesses raised by R4 is missing references to MultiPlaneNeRF, Pix2NeRF, HyperNeRFGAN, and Points2NeRF. We appreciate R4 for bringing this up. However, MultiPlaneNeRF was submitted to arxiv after the submission deadline, Pix2NeRF is already cited in the paper, and while we are happy to add references to HyperNeRFGAN and Points2NeRF, we want to note that they are not yet peer-reviewed. We hope the reviewer reconsiders this unfair criticism of our work.

      - The reviewer's comments on alternative design choices are arbitrary and unsupported by facts (review subsection titled "The model train generalizable NeRF priors"). In our paper, we justify our design choices and provide comprehensive ablations to evaluate them. Unfortunately, the reviewer has listed a number of arbitrary "what about x" design choices with no evidence to suggest that they would work better.

2. **R1: How is the condition retained?**

    - The denoise and finetune step operates on the output of the hypernetwork, $M$, that produces a NeRF, $f$, on a given condition.
    - To retain the condition, we render $f$ to $m$ different poses (as mentioned in line 183). In our experiments, $m=91$.
    - The denoising module improves the texture of these $m$ rendered images. However, it is to be noted that this step only changes the input image marginally (this is also illustrated in the added rebuttal PDF, Figure 3), and thus the condition is not lost.
    - Further, since $f$ is fine-tuned on the denoised images (which are already condition aware), the overall process does not lose the condition.

2. **R3: ... resolving geometric ambiguity through the denoising the 2D outputs does not seem logically coherent**

    - We do not resolve the geometric ambiguities through the denoising process. In fact, our assumption is the exact opposite - the input to the denoise and finetune step should already be multi-view consistent.
    - 3D inductive bias is established through the standard volumetric loss proposed in [30] as mentioned in Equation 2 while training the hypernetwork, $M$. The NeRF predicted by $M$ is already multi-view consistent.

3. **R2 & R3: How are the results multi-view consistent and not blurry?**

    - While training a NeRF on multiview images that are highly inconsistent can result in inconsistency and blurry results, but we would like to point out that the multiview inconsistencies in the denoised images in the “Denoise and Finetune” step are negligible.
        - To begin with, our images (to be denoised) are rendered from a fully-fledged NeRF, which is multiview consistent by design. This is evident from the experimental evaluations where it can be seen that HyP-NeRF significantly outperforms (on ABO) or at least is comparable (on SRN) to the baselines on the task of single (or sparse) view image to NeRF synthesis. This NeRF is rendered to 91 different views ($m=91$).
        - Secondly, even though VQVAE2 denoises the renderings, the denoised images are only marginally different from the input images, and therefore the multi-view inconsistencies are negligible (as clearly shown in the PDF Figure 2), unlike Instruct-Nerf2Nerf and StylizedNeRF that modify the images drastically resulting in inconsistencies and in blurry results.
        - Our final trick is - instead of training a NeRF from scratch, only fine-tune the original predicted fully-fledged NeRF that is already very similar in geometry and texture to the denoised images, thus making sure that the final NeRF is not distorted and retains the initial geometric qualities while showcasing improved texture and fine geometry refinements (like smoothened edges).

**Computation Metrics (inference time, parameters, and FLOPS), rebuttal PDF, Table 2**. In the HyP-NeRF inference time, $m$ denotes the number of images used for the denoise and finetune process. Although we have the largest number of parameters (2$\times$ and 61$\times$ compared to PixelNeRF and CodeNeRF, respectively), our FLOPS are significantly lower than both the baselines (26$\times$ and 28$\times$ less compared to PixelNeRF and CodeNeRF, respectively). This is primarily because, given N query points in a scene, the forward pass through the hypernetwork (computationally expensive) happens only once for the scene. Only the NeRF predicted by the hypernetwork (the less computationally expensive part) is run for each query point.

---

### Decision · Program_Chairs · 2023-09-21

**Decision:**

Accept (poster)

**Comment:**

All but one reviewer agree on accepting the paper, citing the good quality of the results, the novelty of predicting the multi-resolution hash encodings and the demonstrated generalizability of the approach. The remaining reviewer cites missing comparisons, but this AC finds that these references are either unpublished or not directly commensurate, and therefore not a reason for rejection. As such, this paper should be accepted.

Nevertheless, the authors are requested to add additional discussions to the related work to compare to any new references that came up that were published before the NeurIPS submission.